# Continuity of Cancer Care: Female Participants’ Report of Healthcare Experiences After Conclusion of Primary Treatment

**DOI:** 10.3390/curroncol32070399

**Published:** 2025-07-11

**Authors:** Mirna Becevic, Garren Powell, Allison B. Anbari, Jane A. McElroy

**Affiliations:** 1Department of Dermatology, University of Missouri, Columbia, MO 65212, USA; becevicm@umsystem.edu; 2School of Medicine, University of Missouri, Columbia, MO 65212, USA; gmp722@umsystem.edu; 3Sinclair School of Nursing, University of Missouri, Columbia, MO 65212, USA; anbaria@umsystem.edu; 4Department of Family and Community Medicine, University of Missouri, Columbia, MO 65212, USA

**Keywords:** oncology, cancer, breast cancer, cancer survivorship, primary care, care continuity

## Abstract

This study highlights female patients’ experiences with cancer care in the U.S., aiming to improve treatment and support. Among 1224 eligible participants, 57 completed the survey. Most had finished their treatment, with 89% completing it as recommended, though 10% stopped early. While nearly 80% consistently saw the same oncologist, only a third continued seeing the same primary care provider. Patients faced challenges such as employment difficulties (26%), affording medication (21%), and medical bills (15%). The findings emphasize the importance of continuity in care and the need for better financial and employment support during and after treatment. Strengths include consistent oncologist involvement, but gaps in primary care and financial burdens need further attention to enhance patient-centered cancer care.

## 1. Introduction

Cancer burden is expanding globally, remaining the leading cause of death among women [1]. In 2022, approximately 20 million new cancer cases were diagnosed worldwide, with over 53 million individuals living within five years of their diagnosis, highlighting a substantial and growing need for comprehensive survivorship care [2]. Among women, breast, lung, and colorectal cancer are the more frequently diagnosed cancers in the United States but the distribution changes when describing prevalent cases [3]. For females, the top three cancers for prevalence are breast, uterine corpus and thyroid [2,3]. Advances in early detection, surgical interventions, and targeted therapies have considerably improved survival rates, yet the complexity of care coordination during and after treatment continues to challenge healthcare delivery systems [3,4]. In addition, the number of cancer survivors is projected to rise to 22.5 million by 2032, impacting access to care and resources in both oncology and primary care settings [2].

Typically, after a cancer diagnosis, patients primarily interact with oncology specialists who manage the intensive phases of treatment such as chemotherapy, radiation therapy, and surgery [5]. However, continuity of primary care during cancer treatment has emerged as a topic of significant interest and importance for long-term well-being and improved patient outcomes. Evidence indicates that ongoing engagement with primary care clinicians (PCCs) throughout cancer treatment may support comprehensive health management, addressing co-morbidities, psychosocial needs, and preventive care measures that oncology specialists might not fully cover [6,7,8]. Despite these potential benefits, patterns of primary care utilization during active cancer treatment vary widely, and clear guidance on optimal integration remains inconsistent [9,10]. Furthermore, transitioning care back to PCCs post-treatment represents another critical juncture, often associated with fragmented care and uncertainty regarding roles and responsibilities between oncology and primary care physicians [11].

Previous literature has highlighted conflicting viewpoints regarding the optimal models for cancer survivorship care. Traditional oncologist-led models offer specialist expertise but often lack sustainability due to resource constraints [5]. Primary care-led models enhance community accessibility yet may lack oncology-specific expertise [7,12]. In contrast, shared-care models, involving both oncology and primary care providers, are increasingly favored for their balance of expertise and accessibility [6,8]. Recent evidence from stratified care pathways in countries like the UK and Australia indicates improved efficiency and patient satisfaction, suggesting tailored care pathways based on patient risk and need may optimize survivorship outcomes [12,13].

Patients experienced improved cancer care with earlier diagnosis, respectful treatment and overall communication with healthcare providers [14,15]. There are several universal factors that receive high scores from patients regarding the cancer care quality, regardless of patient location or country of origin: receiving timely care, clear and understandable information and communication about their diagnosis and treatment, and respectful care [14]. Several recent studies also found that there is a gap between the care patients receive and the evidence-based recommended care [14,16].

Empirical studies initially raised concerns about shifting survivorship care away from specialists, particularly regarding recurrence detection and patient satisfaction. However, extensive research has demonstrated equivalent or superior outcomes from PCC-led or shared-care models compared to specialist-led care in terms of cancer recurrence, patient satisfaction, and quality of life [9,10]. Despite robust evidence supporting alternative care models, widespread implementation remained limited due to systemic barriers such as poor communication between oncology and primary care providers, unclear delineation of responsibilities, and variability in provider preparedness for survivorship care [12,13].

The 2005 Institute of Medicine (IOM) recommendations for individualized Survivorship care Plans (SCP) upon discharge from cancer treatment have been met with mixed results [17,18]. Despite recommendations from cancer organizations, SCPs are not routinely provided by most cancer centers to PCPs, and recent studies have failed to find consistent evidence that supports the use of SCP in improving patient outcomes [18,19,20]. However, PCPs that have used SCP with their cancer survivor patient populations report their value and utility; however, they have indicated challenges with SCP integration and delivery [17]. Chan et al. found that barriers with implementation of cancer survivorship care models in primary care settings include poor care coordination and limited resources, but found these models helpful in survivor engagement and planning [21].

Patient preferences, however, further underscore the significance of integrated care models. Survivors frequently prefer shared-care arrangements for cancer-specific follow-up while favoring PCP-led management for general healthcare needs [22]. Misalignments between preferred and actual care models highlight the importance of enhanced education, communication, and tailored transition planning to improve survivors’ confidence in primary care-led approaches [11,12].

This study aimed to explore female cancer survivors’ experiences and perceptions regarding continuity of care throughout and after primary cancer treatment. Specifically, the research sought to clarify the role and perceived value of primary care, identify barriers to continuity, and evaluate interactions across care settings. By addressing existing gaps in understanding the optimal integration of oncology and primary care, this study contributes valuable insights to inform clinical guidelines and healthcare policy, ultimately aiming to improve patient experiences and outcomes. Principal conclusions from this research emphasize the necessity for improved coordination and ongoing engagement between oncology and primary care to enhance patient-centered survivorship outcomes.

## 2. Materials and Methods

This was an observational cross-sectional study, approved by the University of Missouri (MU) Institutional Review Board (IRB), number 2096003. Informed consent was obtained from all subjects involved in the study.

Data was collected from 7 May 2024 to 25 June 2024. We utilized ResearchMatch (https://www.researchmatch.org/), a non-profit program supported by the U.S. National Institutes of Health as part of the Clinical Translational Science Award (CTSA) program connecting over 120,000 volunteers interested in research participation. Using the ResearchMatch, our eligibility criteria was 40 years or older (adult cancer diagnosis), female, and treated for cancer diagnosis in the United States. Registry participants were sent a one-time invitation email with a brief description of the study as the recruitment strategy. In the invitation email, a link was given for interested registry participant to access forms created in REDCap. REDCap, (Research Electronic Data Capture) is a secure, web-based application designed for building and managing data collection forms [23]. It includes advanced features such as branching logic that allows for more efficient data collection for participants. The participants read about the consent form and if they were willing, they completed the survey. Finally, if interested in receiving compensation for completing the survey, the participants completed a form capturing their name and address. Information from this form was not used in any analysis. The survey asked demographics information including state of residence, birth year/month, sexual orientation, gender identity, income, educational attainment, race, marital status, employment status and insurance status. A discrimination instrument and specific questions about acceptance of sexual and gender minority patients in the healthcare setting that was asked only of those who identified as sexual and gender minority [24,25]. Participants identified as currently in treatment, follow-up care defined as within 5 years of survivorship after completing primary treatment (of surgery, chemotherapy and/or radiation therapy), or surveillance care defined as 6 or more years after completing primary treatment. Each group was asked about up to 3 cancer diagnoses and included month and year of cancer diagnoses, type, stage, second opinion, treatment info, travel distance to treatment center, frequency of seeing specialists and primary care clinicians and consistency of seeing these clinicians. A list of practical problems, treatment related, and family problems, tailored from the distress thermometer was asked [26]. For those in follow-up or surveillance care, a list of medical and everyday problems related to the cancer diagnosis was also asked [27]. Finally, questions about general satisfaction about treatment experience and use of support groups were queried.

The survey was sent to 1224 eligible participants enrolled in this registry. A total of 64 women responded to the survey, of which 57 completed the survey. Only complete responses were included in this study. A $20 e-gift card was offered to the first 50 volunteers who completed the survey.

Survey responses were analyzed using descriptive statistics to summarize participant characteristics and cancer treatment experience. Frequencies and percentages were calculated for categorical variables overall and stratified by current treatment status (yes/no). For each item, the total number of respondents (*n*), percentage of the full sample (%), row percentage (distribution within each category) and column percentages were calculated. No inferential tests were conducted given the small sample size; only descriptive comparisons are reported.

## 3. Results

Our findings can be grouped into four major categories: type and stage of cancer at diagnosis, distance to specialty care and treatment facilities, surveillance experience, and daily challenges during active treatment.

### 3.1. Cancer Types and Stage at Diagnosis

Most respondents in the study (*n* = 26, 45.6%) were diagnosed with breast cancer, followed by lung (*n* = 13, 22.8%) and uterine (*n* = 3, 5.2%) (Table 1). Of the 57 respondents, 68.4% (*n* = 39) were not currently in treatment, and 31.6% (*n* = 18) were still receiving cancer treatment. Four respondents (7%) did not complete, while 35 (61%) completed the recommended cancer treatment. Most of the respondents in the study were diagnosed at stage I (*n* = 29, 50.9%), followed by stage 0 (*n* = 8, 14.4%), stage II (*n* = 13, 22.8%, and stage III (*n* = 5, 8.8%). One respondent was diagnosed at stage 4 (1.8%), and one respondent was not sure of stage at the time of diagnosis (1.8%).

### 3.2. Travel Distance to Cancer Treatment

Almost 50% of the respondents (*n* = 28) traveled between 10 and 24 miles to see cancer specialists who treated them for cancer, and only 18% (*n* = 31) traveled less than 10 miles. Two respondents reported having to travel between 50 and 74, or over 100 miles, respectively, to see their oncologist. Two respondents (3.5%) traveled 75–100 miles for their care. Of those respondents who are currently in treatment, 46.4% (*n* = 13) travel 10–24 miles, 22.2% (*n* = 4) travel less than 10 miles, and 14.3% (*n* = 1) travel 25–49 miles to receive care.

### 3.3. Consistency with Specialist and Primary Care Services During Cancer Treatment

Most respondents report seeing the same oncologists during their primary cancer treatment (*n* = 50, 87.7%), while 7 respondents (12.3%) reported changing clinicians during this time. In addition, most respondents reported seeing the same oncologist during their primary treatment every time (*n* = 36, 63.0%), 6 respondents (10.5%) reported seeing the same oncologist only occasionally, and 3 (5.3%) reported seeing the same oncologist only rarely.

Most respondents in the study also reported seeing their primary care clinician about every other month (*n* = 11, 28.2%), 8 (20.5%) respondents reported having monthly visits with their PCC, while 9 (23%) had primary care appointments 2–3 times a year, during their cancer treatment (Table 2). Over 50% of respondents reported not having the same PCC as they did at the time of diagnosis.

### 3.4. Follow-Up and Surveillance

Of the 39 respondents not currently receiving treatment, over half (51.3%, *n* = 20) were seeing their PCCs during their follow-up care (<5 years since diagnosis), compared to 41.0% (*n* = 16) during surveillance (>5 years since diagnosis). Thirteen respondents (33.3%) were still seeing their oncologist during follow-up care, compared to 4 (10.3%) during surveillance. Ten respondents (25.6%) were seeing a radiology oncologist during follow-up care, while no respondents reported seeing radiology oncologists in surveillance.

### 3.5. Reported Everyday Problems During Cancer Treatment

Over 25% of respondents in the study (*n* = 10) reported concerns with employment ability or returning to work. Eight (20.5%) of respondents were concerned about their ability to afford medications, and 6 (15.4%) about paying medical bills. Other respondents reported concerns about paying for travel and lodging needed for monthly medical appointments (*n* = 7, 18.0%), meeting monthly financial needs of their households (*n* = 5, 12.8%), having adequate insurance or ability to afford insurance premiums (*n* = 4, 10.3%). Other respondents reported problems with attending school (*n* = 2, 5.1%), having too many medical visits (*n* = 3, 7.3%), and having too few medical visits (*n* = 2, 5.1%). Two respondents (5.1%) reported a lack of clinician-to-clinician communication.

## 4. Discussion

We observed cancer type trends in our study population similar to global cancer trends in women [1]. Nearly half of our respondents were diagnosed with breast cancer, which is the most commonly diagnosed cancer among women in the US and worldwide [1,28]. Recent studies on the global burden of cancer in women showed lung cancer incidence and mortality rates, also second highest cancer type in our study, are highest in North America [1,28].

Only four respondents in our study traveled less than 10 miles to see their oncologist; all the other patients reported over 10-mile travel for their oncology care. Most respondents (70%) traveled more than 20 miles for their care. Previous studies found that travel burden of 20 miles or more prevents most patients from receiving medical care, even in cases where preventative care or other healthcare services offered are free [29]. This burden was also highlighted in everyday problems indicated by respondents, including challenges with paying for travel and lodging needed for specialty appointments. These findings support previous studies which reported travel related to oncology treatment is distressing, tiring, and a major difficulty for patients and families [30].

Many of our respondents were still seeing their oncologist during the follow-up and surveillance care (up to 5 years post primary treatment (follow-up), and more than 5 years (surveillance), indicating that their survivorship care was likely being managed, at least partly, by oncologists. This is consistent with recent longitudinal findings from a cohort of breast cancer survivors (*n* = 1412) where only 20.5% reported shared oncology-PCC survivorship care and 22.4% reported PCC-led survivorship care [22]. Our study also found most respondents had the same oncologist but not the same PCC during their cancer treatment. The role of PCC in cancer care is expanding, and many PCCs are finding shared care models with oncologists of utmost importance, especially during the transition time to follow-up care [12,13,31]. Previous studies emphasized the importance of having an ongoing relationship with PCC or primary care teams and found recurring criticisms with lack of personal continuity [32]. Shared care between oncologists and PCCs, however, may be preferred by PCCs with the caveat that many PCC indicate low self-efficacy in providing comprehensive survivorship care [33]. and the desire for clear communication with their patients’ oncology team. There are, however, opportunities for cancer survivorship education and training for PCCs, using different modalities such as self-directed online education, workshops, webinars, and tele-mentoring sessions [34,35,36].

Which specialist, the oncologist or the primary care clinician, should be primarily responsible for shepherding patients through healthcare decisions and monitoring during the cancer follow-up and surveillance periods has been evaluated for over 25 years. The landmark randomized control study conducted in England found PCC-led care was an acceptable alternative to hospital-based survivorship care for clinical outcomes [37]. Several systematic reviews have subsequently reported similar findings as the landmark study [38,39,40,41,42,43,44,45]. For example, one recent randomized control study in the Netherlands of patients who underwent surgical treatment for colorectal cancer evaluated patients’ quality of life five years post primary treatment between surgeon-led care vs. primary care-led care [46]. They found no clinical difference in quality of life of the participants. However for PCC-led follow-up care to be effective, a strong relationship is helpful and our study found only about half of the participants having the same PCC over time. Further research is needed to explore how long-standing versus newly established relationships with PCCs influence the quality and outcomes of follow-up cancer care.

One aspect of care that has received increasing attention is the financial burden and its effect on the individual, family and ultimately the long-term health of the patient. This financial burden manifests as inability to return to paid employment, inability to afford costs associated with medical services, and insurance concerns as noted in our survey respondents. This well-described economic burden associated with cancer care is termed financial toxicity [47]. For example, as many as 73% of individuals diagnosed with cancer have experienced subjective financial hardship, while nearly half of survivors (around 48%) continue to face monetary challenges [48]. Patients who reported a lot of financial problems were more than four times less likely to report their quality of life as excellent, very good or good [49]. In another study, patients with greater financial issues reported higher anxiety, fatigue and lower social functioning than patients with less financial concerns [49,50,51]. These findings on survivors’ health reflect an objective financial burden faced by many and the need for interventions to mitigate financial toxicity.

A limitation of this study was only 57 participants completed the survey and therefore the results should be viewed with caution; however, our findings were consistent with other reports. Given the heterogeneity of cancer diagnoses within our sample, the applicability of these findings to individuals with different cancer histories may be limited.

## 5. Conclusions

The practical implications of our findings suggest that continuous engagement with primary care clinicians during cancer treatment may improve respondent-reported experiences and highlight areas for enhancing communication and coordination across multidisciplinary teams. Continuity of primary care during cancer treatment could then later facilitate primary care involvement in cancer survivorship. Engagement between PCC and oncology becomes increasingly important as the number of long-term cancer survivors continues to rise, meaning survivorship care becomes more prominent, and ideally, routine. PCPs report willingness to provide cancer survivorship care, but may benefit from additional education and training on specific survivorship care issues [52,53,54,55]. Future research should continue to identify key components of what cancer survivors deem important in their post-cancer healthcare. Gaining perspectives and experiences through qualitative studies with in-depth interviews of cancer survivors, PCCs and oncologists can provide insights into strategies to improve survivorship care. These components should then be incorporated, as possible, into survivorship care models and initiatives that could potentially be facilitated by PCCs.

## Figures and Tables

**Table 1 curroncol-32-00399-t001:** Respondent reported cancer types.

Type	Number of Respondents
Ductal	1
Esophagus	1
Stomach	1
Thyroid	1
Urethral	1
Cervical	2
Colorectal	3
Uterine	3
Not reported	5
Lung	13
Breast	26
Total	57

**Table 2 curroncol-32-00399-t002:** Frequency of PCC visits during cancer treatment.

Frequency	Number of Times	Percent (%)
At least monthly	8	20.5
About every other month	11	28.2
About 4 times a year	6	15.4
About 2–3 times a year	9	23.1
Once a year	2	5.1
Less than once a year	3	7.7

## Data Availability

The raw data supporting the conclusions of this article will be made available by the authors on request.

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
