# Peer review of "Continuity of Cancer Care: Female Participants’ Report of Healthcare Experiences After Conclusion of Primary Treatment"

_curroncol, 2025, doi:10.3390/curroncol32070399_

Round 1

Reviewer 1 Report

Comments and Suggestions for Authors

Thank you for your invitation to review this manuscript. This study identified strengths and challenges in cancer care. Consistent oncologist involvement and proximity to care centers was consistently reported during active treatment. However, several aspects require improvement to enhance the scientific significance. First, the sample size (n=57) is quite limited, which affects the generalizability of findings. In discussion section the authors would be better explicitly acknowledge this limitation and interpret results with greater caution. Second, from my personal point of view, the discussion restates results but does not adequately compare them with previous studies on patient-reported continuity, especially in oncology. At last,  the revision of English language throughout the manuscript could improve the readability of this study.

Author Response

First, the sample size (n=57) is quite limited, which affects the generalizability of findings. In discussion section the authors would be better explicitly acknowledge this limitation and interpret results with greater caution.

Response.  We added this consideration in the abstract:  “This study, albeit for a small number of participants (n=57) identified strengths and challenges in cancer care.” 

In the discussion we also added a limitation sentence, “A limitation of this study was only 57 participants completed the survey and therefore the results sound be viewed with caution and may not be generalizable to all people with a history of cancer.”

Second, from my personal point of view, the discussion restates results but does not adequately compare them with previous studies on patient-reported continuity, especially in oncology.

Similar to the comment about readability of the study that we addressed below, we can describe in more detail the cited studies, as this seems the reviewer would prefer this style.   This will lengthen the paper and I believe we have a word limit. Please advise if the editor would like us to describe the cited studies and additional studies that support our statements in detail, with the understanding that this will lengthen the paper.

At last,  the revision of English language throughout the manuscript could improve the readability of this study.

As the co-authors have published over 200 articles, it is not clear what was unclear about the writing.  This is difficult to address given the generic statement.  If the reviewer found the results section difficult to follow, it would have been helpful to identify that section, as we feel this section was the least reader friendly.  One camp supports having this section be number heavy and another camp believes this section should not necessarily be full of numbers.   Other than this section, we are not sure where in the manuscript that our thoughts were unclear.  Please provide some specific details and we would be happy to edit.

Reviewer 2 Report

Comments and Suggestions for Authors

The authors surveyed cancer surviors concerning issues with the quality of care during surviorship. Among the results, many patients did not have access to the same primary care physician that they had during their initial diagnose. Other issues included financial challenges and access to treatment facilities. This is a small but important study to help improve care of the growing group of cancer survivors. I made an important correction in the text that cited an incorrect date for projected cancer survivorship. I also suggested some ideas for future research. See my comments in the attached pdf.

Author Response

Response to editor and reviewer

  1. Please add city and post code in all of the affiliations.

Response:  we added this information (Columbia, 65212) to all the affiliations.

1University  of Missouri, Department of Dermatology; Columbia, 65212; becevicm@umsystem.edu

2University of Missouri, School of Medicine; Columbia, 65212;  gmp722@umsystem.edu

3University of Missouri, Sinclair School of Nursing; Columbia, 65212; anbaria@umsystem.edu

4University of Missouri, Department of Family and Community Medicine; Columbia, 65212;  mcelroyja@umsystem.edu

  1. Add ‘respondents’ (line 33)

Response:  Thank you for catching this omission.  We added this.  Over 63% of respondents were not seeing the same primary care clinician as they did when they were first diagnosed. Respondents reported facing challenges with employment and ability to return to work (26%), being able to afford medication (21%), and paying medical bills (15%).”

  1. Correct date from 2023 to 2032 (line 57) 

Response:  we corrected the date.

“In addition, the number of cancer survivors is projected to rise to 22.5 million by 2032, impacting access to care and resources in both oncology and primary care settings [5].”

  1. Suggested additional text “You might mention future qualitative studies with in-depth interviews of cancer survivors. Also, future studies should investigate the attitudes of the PCCs and cancer specialists in improving survivorship care?”

Response:  we added the following to lines 239-241  “Gaining perspectives and experiences through qualitative studies with in-depth interviews of cancer survivors, PCCs and oncologists can provide insights into strategies to improve survivorship care.”

  1. Please revise the statement

Response:  “The raw data supporting the conclusions of this article will be made available by the authors on request.”  This is the statement that fits best for this study.

  1. Use the journal reference style.

Response: we updated the reference style per your guidance.